# Acute Promyelocytic Leukemia: Review of Complications Related to All-Trans Retinoic Acid and Arsenic Trioxide Therapy

**DOI:** 10.3390/cancers16061160

**Published:** 2024-03-15

**Authors:** Alexandra Ghiaur, Cristina Doran, Mihnea-Alexandru Gaman, Bogdan Ionescu, Aurelia Tatic, Mihaela Cirstea, Maria Camelia Stancioaica, Roxana Hirjan, Daniel Coriu

**Affiliations:** 1Department of Hematology and Bone Marrow Transplantation, Fundeni Clinical Institute, 022338 Bucharest, Romaniamihnea-alexandru.gaman@drd.umfcd.ro (M.-A.G.); ionescu.bogdan44@yahoo.com (B.I.); auratatic@yahoo.com (A.T.); mihaela.cirstea@umfcd.ro (M.C.); stancioaicamariacamelia@yahoo.com (M.C.S.); hirjan.roxana@gmail.com (R.H.); daniel_coriu@yahoo.com (D.C.); 2Faculty of Medicine, “Carol Davila” University of Medicine and Pharmacy, 050474 Bucharest, Romania

**Keywords:** acute promyelocytic leukemia, arsenic trioxide, all-trans retinoic acid, non-hematologic toxicities

## Abstract

**Simple Summary:**

Acute promyelocytic leukemia (APL) stands as a remarkable success in achieving high cure rates with chemotherapy-free treatment regimens. Despite these remarkable outcomes, hematologists managing patients with APL take into consideration the distinct side effects associated with the two dual differentiation agents, i.e., all-transretinoic acid (ATRA) and arsenic trioxide (ATO). The objective of this paper is to review relevant literature data and discuss the recommended management strategies of non-hematologic early and late complications that arosein patients diagnosed with APL who were treated with ATO and ATRA-based regimens.

**Abstract:**

The hallmark of acute promyelocytic leukemia (APL) is the presence of the characteristic fusion transcript of the promyelocytic leukemia gene with the retinoic acid receptor α gene (PML::RARA). The PML::RARA fusion is a molecular target for all-trans retinoic acid (ATRA) and arsenic trioxide (ATO). Therapies based on ATRA plus ATO have excellent outcomes in terms of complete remission rates, overall survival, and achievement of deep and durable molecular responses with a very low incidence of relapse. However, although the combination of ATRA and ATO has lower hematologic toxicity than standard chemotherapy, its use is associated with a spectrum of distinctive toxicities, such as differentiation syndrome, liver toxicity, QT interval prolongation, and neurotoxicity. Rigorous monitoring of patients’ clinical evolution is indispensable for identifying and addressing each complication. The objective is to maintain an equilibrium between treatment-induced adverse events and therapeutic efficacy. This paper focused on non-hematologic complications associated with the combination of ATRA and ATO. Additionally, we discuss late-onset complications of this therapy. In summary, the majority of treatment-related adverse events are manageable, self-limiting, and reversible. More so, there seems to be a lower incidence rate of secondary neoplasms compared to standard chemotherapy. However, further research is required to assess how the ATRA plus ATO regimen affects the emergence of additional comorbidities.

## 1. Introduction

Acute promyelocytic leukemia (APL) accounts for 10% of all cases of acute myeloid leukemia (AML) [1]. This specific subtype of acute leukemia is distinguished by a high potential for cure using regimens that involve differentiating agents, often requiring minimal or no chemotherapy. Nevertheless, the induction course is marked by a notable incidence of early mortality. The established approach for patients diagnosed with APL involves four essential components: early diagnosis and prompt administration of ATRA; mitigation of early mortality; recognition and management of complications associated with arsenic trioxide (As_2_O_3_, ATO) and all-trans retinoic acid (ATRA) treatment; and monitoring of measurable residual disease (MRD) [2]. Considering the outstanding outcomes of this treatment regimen, with a cure rate exceeding 90% for APL patients, our current main objectives are to reduce the risk of early death, minimize treatment-related toxicities, and enhance patients’ adherence to the prescribed treatment [3,4,5,6,7].

The treatment landscape in APL has undergone significant developments over the years. Initially, anthracycline monotherapy was employed in 1973 by Bernard et al., achieving a 50% rate of complete remission [7,8]. Subsequently, in 1980, ATRA was discovered to induce differentiation of atypical promyelocytes, marking pivotal progress in this field. However, several adverse events began to be noted at that time, e.g., dryness of the lips and skin, digestive symptoms, and headache [7,9]. Another crucial milestone occurred in 1997 with the introduction of combination therapy involving ATRA and chemotherapy, resulting in high rates of complete remission and sustained responses. By 2004, the standard of care for non-high-risk APL had evolved to incorporate a regimen comprising both ATRA and ATO. Starting in 2010, the role of gemtuzumab ozogamicin (GO) in combination with ATRA and ATO for high-risk APL cases began to be investigated. Subsequently, from 2013 onwards, discussions have focused on the use of oral arsenic trioxide [7]. A real-world analysis conducted by Zhu et al. which included patients with APL treated with ATRA (25 mg/m^2^) and ATO (0.15 mg/kg/d) or an oral arsenic formulation, i.e., Realgar/Indigo naturalis formula (RIF), demonstrated a reduced early death rate of 8.2% and a notable survival rate, with a three-year overall survival reaching 87.9% [10].

ATO has a long and intriguing course in medicine, spanning from its historical uses as a toxic substance during the Middle Ages, including the purported poisoning of Napoleon through arsenic-contaminated wine, to its contemporary utilization as a targeted therapeutic agent with exceptional efficacy in the treatment of APL [3]. During the Middle Ages and the Renaissance, ATO was administered as a treatment for diseases such as syphilis and trypanosomiasis [4]. More recently, in the 1970s and 1980s, Chinese researchers pioneered the use of ATO (10 mg/day intravenous infusion) as a treatment for relapsed APL, with complete remission rates ranging from 65.5% to 84% and a survival rate of over 10 years in nine patients [4]. In 2000 and 2001, the United States Food and Drug Administration (FDA) and the European Medicines Agency (EMA) approved ATO as a treatment for relapsed or refractory APL [11]. Later on, the FDA (2018) and EMA (2016) approved the use of the ATO-ATRA combination as a front-line therapy for non-high-risk APL based on the results of the APL0406 phase III randomized multicenter trial published by LoCoco et al. [5].

Since its first use in 1980, ATO has demonstrated mild hematologic toxicity [6]. Furthermore, this drug has a dose-dependent effect, i.e., it induces apoptosis at higher concentrations (0.5 to 2 µM) and differentiation at lower concentrations (0.1 to 0.5 µM), with both scenarios being associated with PML::RARA degradation [7]. The synergistic mechanisms of ATRA and ATO trigger differentiation, apoptosis, and the degradation of PML::RARA [12]. Moreover, in vitro studies have demonstrated that ATO may enhance the effect of paclitaxel, thereby being a promising approach for the management of solid malignancies, e.g., refractory breast cancer [13]. Clinical trials have highlighted the particular early side effects associated with ATO therapy, such as liver impairment, leukocytosis, differentiation syndrome, prolongation of the QTc interval, skin toxicity, and neurotoxicity, most of which occur during induction therapy [5,14]. Regimens containing ATO and ATRA have completely changed the clinical course of patients with APL, leading to high rates of overall survival, extremely low rates of relapse, and successful cures for the majority of patients [5,14,15].

The pivotal APL0406 trial, along with its extended follow-up demonstrated that the treatment regimen comprising ATRA and ATO is significantly superior to standard chemotherapy regarding both event-free survival (97% vs. 86%, *p* = 0.02) and overall survival (with a 2-year overall survival probability of 99% and 91%, respectively) for patients diagnosed with non-high-risk APL [5,15]. Regarding the toxicity profile, it was observed that subjects who received the ATRA plus ATO combination experienced a reduced rate of infections compared to those treated with ATRA plus chemotherapy (*n* = 26 vs. *n* = 59, *p* < 0.001). Furthermore, the combination of ATRA and ATO often led to a reduced need for transfusions. Nevertheless, it is important to highlight that individuals receiving ATO + ATRA exhibited a higher incidence of high-grade hepatotoxicity compared to those who were prescribed ATRA + chemotherapy (63% vs. 6%, *p* < 0.001), along with a higher occurrence of QTc prolongation (16% vs. 0%, *p* < 0.001), respectively [5].

Another representative study, the AML17 trial, enrolled patients diagnosed with both low-risk and high-risk APL. In this investigation, a different dosage regimen of ATO was administered alongside a standard dose of ATRA to all patients included in the ATRA plus ATO arm. Additionally, subjects identified as having high-risk APL received an additional dose of GO as part of their treatment protocol. The study demonstrated that patients treated with ATRA and ATO experienced reduced hospitalization durations and required fewer blood products and antibiotics compared to the ATRA and chemotherapy group. However, the incidence of hyperbilirubinemia of any grade was higher in the ATRA and idarubicin group, while higher levels of aspartate transaminase (AST) were observed in the ATRA plus ATO group [14]. The incidence of cardiac side effects was higher in the ATRA plus ATO arm compared with standard chemotherapy. After two treatment cycles, 11% of patients in the ATO and ATRA group experienced cardiovascular complications, while no such events were observed in the ATRA and idarubicin group (11% vs. 0%, *p* = 0.001) [14]. Moreover, other non-hematological side effects, including elevated creatinine levels, proteinuria, alopecia, nausea, hematuria, and diarrhea, were more common with the ATRA and idarubicin regimen when compared to the chemo-free regimen [14].

Other prospective studies involving patients diagnosed with APL explored the interest of first-line treatment with an ATO plus ATRA-based regimen, especially for high-risk patients. The APL15 study demonstrated that ATRA-ATO without chemotherapy could serve as an effective and well-tolerated treatment for APL patients with all aspects of risk. In this investigation, the cytoreductive strategy employed was based on hydroxyurea, which was titrated up to a maximum dosage of 0.1 g/kg/day. Furthermore, the study incorporated the use of mannitol to increase the permeability of arsenic into the cerebrospinal fluid, aiming to lower the risk of CNS recurrence [16].

Table 1 and Table 2 present a summary of the key studies conducted in APL management, highlighting specific ATRA and ATO treatment schedules, along with incidences of common toxicities such as liver toxicity and cardiac issues, as well as rare complications (see Table 1 and Table 2). Management of grade 3–4 treatment-related toxicities involves temporary discontinuation of ATRA, ATO, or both, followed by dose reduction, with rigorous patient monitoring [5,14].

## 2. Liver Toxicity

Randomized clinical trials have shown that hepatotoxicity is more frequent in patients treated with ATO and ATRA regimens compared with ATRA and conventional chemotherapy [5,14,15,20]. In numerous studies, the assessment of liver impairment involved laboratory evaluations, including measurement of alanine aminotransferase (ALT) and AST levels, along with gamma-glutamyl transferase (GGT), alkaline phosphatase, and total bilirubin concentrations. In most cases, hepatic complications consisted of higher levels of ALT and AST.

Lo Coco et al. demonstrated that liver function abnormalities occurred in 44% of patients who received an ATRA-ATO combination, with 63% of patients experiencing grade 3–4 hepatic side effects, in contrast to 6% of patients treated with ATRA and chemotherapy (*p* < 0.001) [5]. The hepatic toxicity rates reported by Burnett et al. were 25% among patients treated with ATO plus ATRA, suggesting a comparatively lower incidence, possibly associated with the distinct schedule of ATO administration. In the AML17 study, the use of GO as a cytoreduction agent for hyperleukocytosis was not linked to an elevated incidence of hepatotoxicity [14,21]. In the APML4 trial, which included 124 patients who received ATRA, ATO, and idarubicin, 44% developed grade 3–4 liver toxicity during the induction phase, contrasting with a minimal 2% occurrence during consolidation [17,18]. Table 1 provides a summary of the occurrence of liver toxicity observed in various clinical trials or real-world studies, focusing on different administration schedules of the ATRA + ATO regimen.

Generally, liver abnormalities predominantly manifest during the induction phase, with a notably lower incidence observed during consolidation [5,14,17]. Different trials have delineated possible predictive factors for ATO-induced hepatotoxicity, such as hemoglobin levels ≥ 80 g/L, the absence of prophylactic hepatoprotective agents, non-single-agent ATO, fibrinogen concentrations < 1 g/L [23], and the presence of homozygous mutation of MTHFR 1298 (C/C) [24]. However, further studies are necessary to establish the specific contribution of each of these factors to the occurrence of liver toxicity. Moreover, evidence suggests that differentiation syndrome may serve both as a risk factor and as an exacerbating factor for hepatotoxicity [25].

ATO-induced hepatotoxicity is a diagnosis of exclusion; therefore, other etiologies must be considered, including reactivation of several viral infections, e.g., hepatitis B, hepatitis C, or cytomegalovirus (CMV) infection reactivation. Imaging studies must be performed to exclude local complications such as cholecystitis or local thrombosis, and the identification of all other medications that could potentially contribute to liver toxicity is necessary.

The management strategy needs to be adapted to the severity of the manifestations. In patients with grade 1–2liverfunction abnormalities, the same dose of ATO may be maintained with closer hepatic monitoring. In the context of grade 3–4 hepatotoxicity, a necessary step is the temporary discontinuation of ATO. Subsequently, ATO can be resumed at 50% of the initial dose for 7 days, provided that liver enzyme levels decrease to less than 3 times the upper limit of normal (ULN) or that the total bilirubin level decreases to less than 1.5 times the ULN. In the absence of any increase in liver markers during these 7 days, ATRA and/or ATO can be reintroduced at their full dosages [2,5].

Routine surveillance of liver function tests is recommended for patients undergoing treatment with ATO and ATRA. Finally, various clinical trials reported that liver toxicity is transient and does not lead to liver failure or permanent treatment discontinuation [5,14,25,26].

## 3. QT Prolongation

Treatment with ATO can lead to various cardiac toxicities, particularly during induction. These adverse cardiac events include QTc prolongation and other conduction disturbances, recurrent tachycardia, T-wave inversion, and left ventricular systolic dysfunction [5,15,17,18]. It is worth noting that, despite the frequent occurrence of QT prolongation, it lacks clinical significance in the majority of cases. The exact mechanism responsible for ATO-induced cardiotoxicity remains largely unknown. It might be secondary to increased oxidative stress and intracellular calcium overload. Several ongoing studies are investigating the potential of antioxidant and anti-inflammatory agents such as sacubitril/valsartan, resveratrol, L-ascorbic acid, and omega-3 fatty acids as potential therapeutic options to attenuate arsenic-induced cardiotoxicity [25,27,28,29]. Further research is needed to fully investigate and validate the efficacy of these treatments.

QTc prolongation is defined as a QTc of more than 450 milliseconds in men and 460 milliseconds in women, with the correction calculated according to the Framingham formula [5]. Alternative rate-adjusted formulas, including Fridericia and Hodges, can also be used [2]. Roboz et al. have shown that the use of rate correction methods other than the Bazett formula significantly reduces the rate of estimated QT values greater than 500 ms. This approach has been shown to lead to fewer withheld or omitted doses during ATO treatment [2,30].

Prolongation of the QT interval represents a major risk factor for the development of torsades de pointes (TdP) and sudden cardiac death. Other factors associated with TdP include electrolyte abnormalities (such as hypokalemia and hypomagnesemia), female gender, pre-existing cardiovascular conditions (e.g., heart failure, bradycardia), drug interactions, genetic predisposition, and a family history of arrhythmias [31]. The drugs most frequently associated with QT prolongation include antiemetics (ondansetron, prochlorperazine), antidepressants (including tricyclic antidepressants and selective serotonin reuptake inhibitors), antibiotics (quinolones), antifungals (azoles), potassium-wasting diuretics, and antiarrhythmics (amiodarone, sotalol). Azole antifungals, including itraconazole, voriconazole, and posaconazole, have been linked to QT prolongation. In contrast, isavuconazole is the only azole that does not influence QT prolongation. Furthermore, at certain plasma concentrations, isavuconazole has been demonstrated to have QT-shortening effects [32].

Sanz et al. recommend the utilization of telemetered ECG in situations involving patients at very high risk, such as those experiencing significant QT prolongation or TdP with accompanying clinical symptoms such as dizziness and syncope, or those with other risk factors. On the other hand, Roboz et al. propose daily ECG monitoring as a potential option for supervising patients treated with medications that may prolong the QT interval [2,30]. These monitoring approaches are important for early detection and management of potential cardiac arrhythmias, ensuring comprehensive care for at-risk patients.

In the event that the QTc exceeds 500 milliseconds, the first step is to temporarily discontinue treatment with ATO. Maintaining optimal electrolyte levels, especially potassium (>4 mmol/L) and magnesium (>1.8 mg/dL), is of the utmost importance [2,26]. Possible interactions with other medications that are known to prolong the QTc interval must be checked. After normalization of the QTc interval, which according to the European Leukemia Net (ELN) 2019 recommendations should not exceed 460 ms, ATO treatment is reintroduced at a reduced dosage of 0.075 mg/kg (50% of the full dose) for an initial period of 7 days. If no further QTc prolongation occurs during this time, the ATO dose may be increased to 0.11 mg/kg in the following week. If no further prolongation is detected, ATO is then resumed at the originally prescribed full dose. This approach aims to mitigate the risk of QTc-related complications and ensures a gradual reintroduction of ATO with careful monitoring for any adverse cardiac effects [2,5,11,33].

The assessment conducted by LoCoco et al. revealed a QTc prolongation incidence of 16% in patients treated with ATRA-ATO, in contrast to 0% of patients in the ATRA and chemotherapy arm (*p* < 0.001) [5]. Additionally, Platzbecker et al. demonstrated a higher incidence of QTc prolongation in the ATO + ATRA group when compared to chemotherapy + ATRA, with rates of 8.5% vs. 0.7% (*p* = 0.0022) during induction, and 1.5% vs. 0% (*p* = 0.23) during the third consolidation phase, respectively [15]. As documented in the existing literature, the majority of patients experienced asymptomatic QT prolongation [11,14,30]. In the APL0406 trial, ATO had to be permanently discontinued in one patient, due to QT prolongation [5]. Burnett et al. reported that patients treated with ATO and ATRA experienced more cardiac adverse events after two cycles than with chemotherapy (11% vs. 0%, *p* = 0.001) [14] (see Table 1). The study by Soignet et al. reported a single case (1/40) involving a 7-beat run of TdP in the context of hypokalemia, hypomagnesemia, and concomitant administration of other QT-prolonging drugs [11]. Roboz et al. showed that approximately two-thirds of non-APL patients treated with ATO exhibit prolonged QT intervals. Importantly, despite this high incidence, the occurrence of TdP was found to be very low and did lead to an increase in mortality [30]. In the study led by Iland et al. (APML4), 14% of the patients experienced QTc prolongation during treatment, yet none of them progressed to severe arrhythmias. Notably, during the consolidation phase, a single episode of reversible ventricular tachycardia was observed, and the occurrence rate of other cardiac events during induction was only 1% [17].

Outside of clinical trials, isolated cases of transient, non-sustained ventricular tachycardia without QT prolongation have been observed, which were resolved by the administration of nadolol [34]. Notably, an increased incidence of QT prolongation and liver toxicity was observed in individuals with moderate or severe renal impairment [35].

The aforementioned findings collectively highlight the increased incidence of QT prolongation during ATO treatment, as well as the overall low incidence of severe arrhythmias, thereby contributing to a more comprehensive understanding of the cardiac safety profile associated with ATO therapy.

At present, no specific recommendations have been issued regarding cardiovascular assessments during the long-term follow-up of APL patients who have received ATRA and ATO. Nevertheless, patients who have been prescribed anthracyclines and who exhibit multiple cardiovascular risk factors are considered at an elevated risk for cardiovascular dysfunction. From our point of view, the cases above should be monitored during APL long-term follow-up according to the NCCN Guidelines for Survivorship, specifically for assessing the risk of cardiovascular disease [36].

## 4. Differentiation Syndrome

Differentiation syndrome (DS) stands as a leading cause of early mortality in APL. Consequently, recognition of DS signs and symptoms requires a high level of suspicion and immediate initiation of treatment [37]. The reported incidence of DS varies across a spectrum from 2% to 48%, accompanied by an average mortality rate of approximately 1%, as documented in the existing literature [38,39]. The incidence of DS shows significant differences when comparing clinical trial results with the real-life setting. These differences can be attributed to the fact that there is no pathognomonic clinical or biomarker for the diagnosis of DS. To establish this diagnosis, other possible conditions such as sepsis, hemorrhage, and heart failure must be considered and ruled out.

DS can present with a variety of manifestations, with the most common being unexplained fever, dyspnea, hypotension, weight gain > 5 kg, radiographic lung opacities, acute renal failure, and pleural/pericardial effusion [40]. Depending on the severity of its clinical presentation, DS may be classified as indeterminate (one or two signs or symptoms), moderate (three symptoms), or severe (more than four clinical features) [38]. This range of presentations emphasizes the importance of a thorough clinical assessment and early recognition for appropriate management.

The onset of DS can occur anywhere between 0 and46 days after the introduction of ATRA, with a median time of 12 days [41]. In the study by Montesino et al., it was observed that 47% of patients undergoing ATRA and chemotherapy experienced DS within the first week, while 25% developed DS by the third week. However, a small proportion, specifically, 3% of patients, showed symptoms after the 29th day of ATRA initiation [40]. Characteristic features of early severe DS included an increased incidence of pulmonary infiltrates and weight gain, probably related to an initial increase in white blood cell (WBC) counts and fluid overload. In contrast, late severe DS was characterized by a higher prevalence of hypotension, unexplained fever, pericardial effusion, and renal failure [40].

Previous reports have shown an association between a higher incidence of DS and factors such as white blood cell (WBC) counts at diagnosis and peak levels during induction, elevated serum creatinine levels, the APL microgranular subtype, a higher body mass index (BMI), and CD13 and CD11b expression on flowcytometry [40,42,43]. However, there are currently no well-established predictive factors for DS. Leukocytosis occurring during ATO and ATRA induction must be managed with cytoreductive therapy such as hydroxyurea (HU) 500 mg four times/day for WBC < 50.000/mmc and 1 g four times/day for WBC > 50.000/mmc [2,5]. Furthermore, in cases of “extreme leukocytosis” or when resistance to HU is observed, it is recommended to consider the use of idarubicin or GO [37]. Research by La Bella and colleagues illustrated that timely intervention for hyperleukocytosis through the administration of cytoreduction chemotherapy (cytarabine/daunorubicin/idarubicin) has the potential to improve patient outcomes and decrease healthcare-related costs, particularly in patients with non-high-risk APL treated with the ATRA-ATO regimen [44]. Furthermore, De Botton et al. demonstrated a notable increase in the occurrence of DS in the arm where ATRA and chemotherapy were not commenced concurrently, as opposed to the arm where chemotherapy and ATRA were initiated simultaneously, with rates of 18% and 9.2%, respectively (*p* = 0.035) [45].

Several studies support different attitudes regarding the potential benefits of corticosteroid prophylaxis in mitigating the incidence of DS. While some investigations indicate positive effects, others report no significant impact. Various schedules of prophylactic corticosteroids have been proposed, including prednisone at a dosage of 0.5–1 mg/kg per day starting from day 1 until the completion of the induction phase or for the initial 15 days. Alternatively, dexamethasone at 2.5 mg/m^2^ every 12 h during days 1–15 and methylprednisolone at 20–50 mg/day for 5–10 days have been suggested [38,41,46]. In most treatment protocols involving ATRA and ATO, corticosteroid prophylaxis is recommended from day 1 [5,17,21]. However, other sets of criteria for initiating corticosteroid prophylaxis include WBC >5000/mmc and elevated creatinine levels > 1.4 mg/dL [2,5,38].

In the treatment of severe forms of DS, the established protocols involve promptly discontinuing the administration of ATRA and ATO. Additionally, patients are initiated on dexamethasone at a dose of 10 mg twice daily until resolution of symptoms, with the possibility of dose escalation to 10 mg every 6 h if necessary [2,5]. As reported in Montesino’s findings, patients experiencing severe DSrequired a comprehensive treatment approach, which included intravenous dexamethasone, diuretics, dialysis, and mechanical ventilation [40].

In the trial conducted by LoCoco et al., DS occurred in 19% of the patients treated with ATRA-ATO, while 16% of the patients in the ATRA–chemotherapy group developed DS (*p* = 0.62) [5]. The incidence of severe DS was comparable in both treatment arms, but notably, two patients who received ATRA plus chemotherapy experienced fatal outcomes. Table 3 provides details on the prophylaxis and incidence of DS in patients treated with ATO + ATRA-based regimens.

## 5. Neurological Toxicities

The use of ATRA and/or ATO has been associated with several neurological toxicities, including headache, peripheral neuropathy, dizziness, mood alterations, insomnia or somnolence, benign intracranial hypertension or pseudotumorcerebri (PTC), and seizures. The most common neurological toxicity was headache, which occurred in both induction, consolidation, and maintenance regimens, respectively, with a frequency ranging from <1% to 60%. Headache was mostly reported as a grade 1–2 neurological toxicity; however, grades 3–4 cases of headache were highlighted in several manuscripts and some APL patients required temporary treatment discontinuation [11,17,18,47]. Wang et al. have concluded that headache episodes are more likely to develop in subjects treated with ATRA + ATO rather than ATO alone (15.79% vs. 6.30%; RR = 1.96 95% CI 0.95–4.07) [47].

Nevertheless, a potentially life-threatening complication linked to the use of ATRA + ATO is PTC, also known as idiopathic or benign intracranial hypertension. Diagnosis is based on the Dandy criteria: the presence of papilledema, normal neurological exam except for cranial nerve abnormalities, an elevated lumbar puncture opening pressure (≥250 mmHg or ≥280 mmHg for non-obese children), and normal neuroimaging studies [48]. Female sex, age (particularly the childbearing years in women), and endocrine/metabolic disorders (e.g., obesity, hypervitaminosis A) are listed among the risk factors for PTC development [49,50]. It has a high incidence in the pediatric population and young adults, especially within the first 2–3 weeks of APL induction treatment regimens [51]. PTC may present with a wide variety of clinical manifestations, such as persistent and severe headache, diplopia, vomiting, nausea, and/or pulsatile tinnitus [52].

The primary complication of PTC is blindness, caused by the progressive swelling and atrophy of the optic disc. Management strategies primarily aim to reduce intracranial pressure and may involve discontinuation of ATRA and ATO, therapeutic lumbar punctures to remove cerebrospinal fluid (CSF), administration of acetazolamide, corticosteroids, diuretics (mannitol or furosemide), and analgesics [48,52,53]. One factor to take into account is the potential for drug interactions with antifungal azoles used for prophylactic or therapeutic purposes, as well as CYP3A4, CYP2C8, and CYP2C9 inhibitors, which can increase the risk of PTC [54,55]. ATRA-related neurotoxicity also shares similarities with vitamin A intoxication, leading to impaired CSF reabsorption. Conversely, the precise mechanisms involved in ATO-induced neurological toxicity have yet to be clarified [51]. Limited data are available regarding the incidence of PTC in APL subjects undergoing dual differentiation therapy. Smith et al. reported five PTC cases in non-high-risk APL patients, four of whom experienced symptom onset during the induction phase of ATO-ATRA treatment, while one individual developed PTC during consolidation. These cases were effectively managed through the temporary cessation of ATRA and the administration of either acetazolamide or topiramate [56].

Montesino et al. evaluated 1034 APL patients who were enrolled in the LPA96, LPA99, and LPA2005 trials, respectively, highlighting an incidence rate of PTC of 3% (*n* = 32 patients). The majority of cases occurred during the induction phase and only two cases were described during consolidation. PTC seemed to occur mostly in subjects aged <18 years old, with fibrinogen levels <170 mg/dL, and with ECOG (Eastern Cooperative Oncology Group) scores >1 [53]. Nevertheless, Montesino et al. showed that this neurotoxic complication was reversible and did not impact treatment outcomes [53].

Peripheral neuropathy seems to be one of the most frequently encountered neurological adverse events associated with the use of ATRA plus ATO, affecting around 25% of APL patients [15,19,47]. The final analysis of the APL0406 clinical trial concluded that this adverse event is the most common neurological side effect of the ATRA + ATO regimen [5]. According to Wang et al., neuropathy is less likely to occur due to the ATRA + ATO regimen versus ATO monotherapy (0% vs. 5.51%; RR = 0.32, 95% CI 0.04–2.24) [47]. These results were reinforced by Platzbecker et al.’s findings, which demonstrated that the addition of ATO to ATRA is linked to higher rates of neurotoxicity when compared to ATRA alone, particularly during the consolidation phase (4.2–5.9% vs. 0%) [15]. Most cases of peripheral neuropathy were, however, reversible and rated as grades 1–2, with serious adverse events reported only in 2.7–8% of patients [15,19]. Over an extended observation period of 12 years, Zhu et al. reported the absence of neurologic complications [57]. Interestingly, according to Zacholski et al., ATO-related neuropathy seems dose-dependent, with dose capping at 10 mg leading to reduced rates of neurotoxicity during the consolidation phase [52]. Supporting the same concept of dose-dependent neurotoxicity, Loh et al. demonstrated that an induction dose of ATO exceeding 500 mg along with an elevated BMI (obesity class II-III) serve as risk factors for neurological complications [58].

Other neurological side effects associated with the use of ATRA + ATO include seizures, mood swings, or dizziness [17], as well as insomnia [11]; however, these complications have only exceptionally been rated as grade 3–4 adverse events. Table 4 provides a summary of neurological complications arising from various schedules of ATO +/− ATRA +/− GO ATRA use in different clinical trials.

## 6. Musculoskeletal Toxicities

The combination of ATRA and ATO has been linked with the development of musculoskeletal toxicities, including musculoskeletal pain (MSK), myopathy/myositis, rhabdomyolysis, and osteoarthritis/degenerative arthritis (see Table 5). When compared to ATO monotherapy, the addition of ATRA was linked to increased incidence rates of bone pain (5.85% vs. 1.57%; RR = 2.50, 95% CI 0.57–10.87) [47]. A long-term assessment of comorbidities in patients with APL revealed a tendency of individuals who were treated with ATRA + ATO versus ATRA + chemotherapy to experience more back pain (30.1% vs. 28.2%) but less overall pain and fewer cases of osteoarthritis/degenerative arthritis (15.7% vs. 20.5%) [59]. Grades 3–4 musculoskeletal toxicities have only exceptionally been noted; they were detected during induction or first consolidation phases and when ATRA + ATO was combined with the use of anthracyclines [17]. Several case reports indicated that ATRA therapy during induction could induce myositis [60,61], especially in the setting of Sweet’s syndrome. He et al. describe the case of a 68-year-old patient who developed rhabdomyolysis secondary to ATO treatment [62]. Kubiak et al. showed that 21.4% of patients (9/42) developed acute MSK during induction, with a median duration of pain of 5 days and a median time of presentation of 11 days after treatment initiation. Its occurrence seems to be related to the rapid increase in WBC counts in patients with low-risk APL [63]. Despite no standard recommendations for its management, temporary ATO + ATRA discontinuation and simultaneous administration of corticosteroids were successful in alleviating acute musculoskeletal pain syndromes [63].

## 7. Skin Toxicities

Dermatological manifestations following APL treatment are sometimes mentioned as potential adverse events of both chemotherapy and chemotherapy-free regimens. Examples include skin rashes, cutaneous and mucosal dryness (including cheilitis, stomatitis, etc.), Sweet syndrome, and ulcerations of the integument, as well as herpes zoster [14,17,20,47]. APL patients mostly exhibited dermatological side effects during induction chemotherapy, with lower rates being reported during consolidation or maintenance cycles [14,20,47]. Chen et al. found higher rates of skin toxicities when ATRA and ATO were combined with chemotherapy, but only in the high-risk APL subgroup [20].

The use of ATRA + ATO vs. ATO monotherapy in APL seems to be linked to an increased rate of dermatologic reactions (27.49% vs. 13.39%; RR = 2.83, 95% CI 1.41, 5.65) and particularly mouth dryness (7.02% vs. 1.57%; RR = 5.71, 95% CI 1.46, 22.40) [47]. Most side effects were self-limiting or resolved with supportive treatment without the need for dose reductions and/or therapy discontinuation [14,17,20,47]. However, grade 3/4 alopecia has been noted with the use of ATRA/ATO either alone or in combination with anthracyclines. However, we must point out that many assessments do not specifically describe the type of skin reactions that occurred with the use of ATRA and/or ATO; thus, more research must be carried out to accurately shape the landscape of dermatological side effects linked with this treatment regimen [14,17,20].

## 8. Other (Rare) Toxicities

The use of ATRA and/or ATO in APL management has also been associated with toxicities involving mainly the metabolism of lipids and carbohydrates [15,17,18,20,59] (see Table 2). The development of hyperglycemia seems more likely related to the administration of ATO, as the use of ATRA + ATO regimens has been reported to cause lower rates of these metabolic complications vs. ATO monotherapy (1.17% vs. 2.36%; RR = 0.67, 95% CI 0.14–3.17) [47]. Most cases of hyperglycemia, hypertriglyceridemia, and/or hypercholesterolemia were registered as grade 1–2 side effects and did not require dose reductions or treatment discontinuation, only supportive management. However, grade 3–4 hypertriglyceridemia and/or hypercholesterolemia have been described, mostly during the consolidation phase, implying that metabolic side effects of ATRA and/or ATO may be related to the duration of exposure [15,20].

A peculiar adverse event linked to the use of ATRA and/or ATO is the development of treatment-related acute pancreatitis. Although rare, acute pancreatitis can occur in the setting of ATRA-induced hypertriglyceridemia, ATRA-induced DS, or ATO treatment [64]. Nevertheless, a review of the literature identified three cases of acute pancreatitis in patients diagnosed with APL which were linked to ATO administration. The mechanism of ATO-related pancreatitis is not well understood; however, several cases of arsenic intoxication associated with signs of pancreatitis, gastroenteritis, and neurologic manifestations have been reported by several authors [65,66]. Given the limited understanding of the possible implications of this rare complication, we advise vigilant monitoring of amylase and lipase levels during APL induction treatment. This approach may facilitate dose adjustments or, if needed, allow early ATO discontinuation before the onset of life-threatening complications.

## 9. Infectious Complications

Infectious complications occur less frequently in APL patients treated with ATRA and ATO than in the setting of chemotherapy, which is undoubtedly related to the lower rate and shorter duration of treatment-induced neutropenia.

In the APL0406 trial, the incidence of grade 3 or 4 neutropenia was notably reduced in patients undergoing treatment with ATRA-ATO in comparison to those treated with ATRA–chemotherapy, both during the induction phase and across all three consolidation cycles [5]. This was associated with a reduced incidence of infectious episodes, with 26 episodes observed in the ATRA-ATO arm compared to 59 episodes in the ATRA–chemotherapy subgroup (*p* < 0.001) [5].

A study published in 2021 by Autore et al. identified two cases of herpes simplex virus (HSV) reactivation and one case of pneumonia following induction treatment by ATRA and ATO in a series of 23 patients. However, 4of the 23 patients had also received one or two doses of idarubicin, with no separate data available regarding specific complications in this subgroup. During consolidation treatment, which uniformly consisted of ATRA and ATO, two cases of cystitis, two cases of fever of unknown origin, one HSV reactivation, and one case of grade 1 diarrhea were described [67].

In an Italian investigation by Pagano et al., the rate of invasive fungal infections among the 103 APL patients treated with differentiating agents was only 7%, as opposed to 24% in the 881 patients who received standard chemotherapy for AML [68]. Based on these findings, the study advised that mold-active antifungal prophylaxis may not be required as a support treatment in the case of chemotherapy-free regimens.

The APL2012 trial, which randomized patients to ATO + ATRA based-consolidation therapy versus ATRA–chemotherapy, demonstrated a higher prevalence of grade 3–4 infections in individuals assigned to the ATRA–chemotherapy arm. This finding remained consistent regardless of the APL risk category. Specifically, in low-risk subjects, the incidence of grade 3–4 infections was 14.1% in the ATO + ATRA group compared to 30.8% in the ATRA–chemotherapy group (*p* = 0.032). In high-risk cases, the corresponding data were 22.7% in the ATO + ATRA group and 47.8% in the ATRA–chemotherapy group (*p* < 0.0001) [20].

Burnett et al. documented a case in which a patient deviated from the established protocol during induction due to a fungal infection [14]. In Iland et al.’s assessment, most infections (76%) occurred during induction, with the majority being cases of sepsis and/or catheter related (46%). During the consolidation phase, the incidence of infectious complications dropped to 3%. Additionally, during induction, one patient included in the study developed aspergillosis [17].

Some reports have indicated an elevated incidence of herpes zoster in patients undergoing ATO treatment, potentially linked to dysregulation of T-cell immune responses [69]. These data are supported by the findings of Freyer et al., who documented a frequency of 13 out of 112 cases (11.6%) of herpes zoster within 6 months of completing ATO treatment. Notably, this included one case of herpes zoster encephalitis. The study emphasized the importance of antiviral prophylaxis, especially in older patients and in those with a prior history of herpes zoster [70].

The NCCN Guidelines for the Prevention and Treatment of Cancer-Related Infections indicate that patients with acute leukemia are at significant risk of infection. Therefore, during periods of neutropenia, the use of fluoroquinolones may be considered for antibacterial prophylaxis. Additionally, antifungal prophylaxis should be taken into account during neutropenia, with careful consideration of potential drug–drug interactions. Prophylaxis against HSV is recommended throughout all treatment cycles [71]. Currently, treatments involving ATO and ATRA are considered less immunosuppressive. In clinical practice, decisions on antibacterial and antifungal prophylaxis now largely depend on local protocols.

## 10. Late-Onset Side Effects

According to data from the available literature, chemotherapy-free regimens combining ATRA and ATO have a promising long-term toxicity profile. After complications related to the early induction phase have been successfully managed, both relapse rates and non-relapse mortality are very low.

It is widely recognized that using an anthracycline-based treatment carries risks, including cardiotoxicity and the development of secondary cancers. In the study by Montesinos et al., it was shown that for patients undergoing a regimen based on ATRA–chemotherapy, there was a 2.2% six-year cumulative incidence of treatment-related myeloid neoplasms (including treatment-related myelodysplastic syndrome and treatment-related acute leukemia). Furthermore, out of 918 patients, 14 developed secondary solid tumors [72]. In contrast, Kayser et al. demonstrated that, following an average follow-up of 1.99 years, patients receiving ATRA and ATO treatment did not develop any secondary cancers [22].

Abaza et al. conducted a comprehensive assessment of 187 APL cases who had been treated with ATRA + ATO (+GO if hyperleukocytosis was present), of whom 26 subjects experienced death. In their study, the median follow-up was 47.6 months. In total, there were seven deaths noted during the induction cycle and in two cases the patients died due to relapsed/refractory APL. The remaining 17 individuals were in complete remission at their time of death and were considered to have died from unrelated causes, e.g., second primary malignancies (*n* = 8), sepsis (*n* = 3), renal failure (*n* = 2), and cardiac arrest (*n* = 1); the cause of death remained unknown for a total of 3 patients. The median time to death was 1224 days [21].

In their assessment of 112 newly diagnosed patients with APL, Zhu et al. found higher rates of liver dysfunction in patients treated with ATRA and ATO compared to healthy controls (15.2% grade 1 liver dysfunction, *p* < 0.001, 42.9% hepatic steatosis, *p* < 0.001) but no long-term liver fibrosis after a follow-up period of 12 years. Additionally, patients had no common signs of chronic arseniasis, e.g., chronic renal failure, neurological abnormalities, or cardiovascular events [57]. In a study by Shetty et al. involving 91 APL patients treated with ATRA and ATO, it was found that after a follow-up period of at least three years, there was a higher incidence of hypertension (39%), diabetes mellitus (25%), and cardiac diseases (11%) in comparison to those treated with ATRA and chemotherapy. There was no significant difference between the two groups regarding the incidence of second primary malignancies [73].

Concerning the long-term quality of life, Efficace et al.’s study, which included patients from the APL0406 trial, revealed that the subgroup treated with ATRA-ATO demonstrated a higher quality of life compared to those undergoing ATRA–chemotherapy [59].

Abedin and Altman recommend regular monitoring of patients’ liver function after treatment completion, especially in those who have experienced acute therapy-related hepatotoxicity. Additionally, they propose close monitoring for hypertension and diabetes, as well as epidemiologically appropriate screening for potential secondary cancers [30]. Furthermore, an assessment conducted by Yin et al. has revealed that beyond 5 years, APL patients do not have an elevated risk of death from non-cancer-related diseases when compared to the general population [74].

At present, for the cancer follow-up of APL patients undergoing treatment with chemotherapy-free approaches, we recommend age-appropriate screening for potential new primary cancer together with comprehensive evaluations of patients’ family and medical history, detailed physical examinations, and routine blood work.

## 11. Conclusions

In summary, the short-term complications associated with ATO plus ATRA therapy are manageable and reversible in adult patients diagnosed with APL. DS remains an important contributor to early mortality in APL. Goals for APL treatment include preventing/reducing early death, improving supportive care, and preventing and caring for treatment-related toxicities. In light of the significant number of patients cured, comprehensive long-term surveillance is necessary. This should include routine follow-up visits, addressing late-term effects, and identifying health problems unrelated to the APL diagnosis.

## Figures and Tables

**Table 1 cancers-16-01160-t001:** Occurrence of cardiotoxicity and liver toxicity in major clinical trials or real-world cohort studies based on treatment regimens for APL.

Study Details	Treatment Regimen	Cardiotoxicity	Liver Toxicity
APL0406 (2013) [5] Platzbecker et al. (2016) [15] prospective phase 3 multicentric RCT nonHR APL ATRA + ATO (*n* = 77) vs. ATRA + CHT (*n* = 79) Median age: 44.6 yrs	Induction ATRA 0.15 mg/kg/d + ATO 45 mg/m^2^/d Consolidation ATRA + ATO (4 cycles)	Induction - severe QT prolongation on D3 (*n* = 1) → ATO discontinuation - repetitive tachycardia during induction (*n* = 2) - QT prolongation: ATRA + ATO vs. ATRA + CHT = 16% vs. 0% (*p* < 0.001) Consolidation 2nd cons.: 2% (ATRA-ATO) vs. 0% (ATRA + CHT) 3rd cons.: 1.5% (ATRA + ATO) vs. 0% (ATRA + CHT) (*p* = 0.23)	Induction and Consolidation - grade 3–4 DILI: 63% (ATRA + ATO) vs. 6% (ATRA + CHT) (*p* < 0.001) - induction 48% vs. 3rd cons. 3.2%
AML17; Burnett et al. (2015) [14] phase 3 multicentric RCT ATRA/ATO ± GO (*n* = 116) vs. ATRA + CHT (*n* = 119) LR (*n* = 86) and HR (*n* = 30) APL Median age: 47 yrs	Induction ATRA 45 mg/m^2^/d + ATO 0.3 mg/kg D1–5 W1and 0.25 mg/kg twice weekly W2–8 C1 ± GO 6 mg/m^2^ (HR pts + 7 nonHR pts) Consolidation ATRA (2 wks on, 2 wks off schedule × 5 cycles) + ATO 0.3 mg/kg D1–5 W1 and 0.25 mg/kg twice weekly W2–4 of C2–5	Induction - grade 3/4 CV events: 1% ATRA + ATO vs. 5% (grade 3) and 1% (grade4) in ATRA + CHT - drug discontinuation (*n* = 2) Consolidation CV events (grade 3): 3% ATRA + ATO vs. 0% ATRA + CHT post-C2: CV toxicity: 11% vs. 0% (*p* = 0.001)	Induction grade 3 elevated ALT: 20% (ATRA + ATO) vs. 8% (ATRA + CHT); Consolidation DILI: similar in ATRA + ATO vs. ATRA + CHT
APML4; Iland et al. (2012) [17,18] non-randomized phase 2 multicentric trial *n* = 124 pts (HR: *n* = 23) Median age: 44 yrs	Induction ATRA 45 mg/m^2^/d D1–36 + IDA 6–12 mg/m^2^ D2,4,6,8 + ATO 0.15 mg/kg D9–36 Consolidation (2 cycles) ATO + ATRA Maintenance (2 yrs) ATRA + 6-MP + MTX	Induction - persistent deep T-wave inversion → study withdrawal (*n* = 1) - QT prolongation (>500 ms): 14%; reversible, no clinical impact - other CV events: conduction abnormalities, LV systolic dysfunction (1%) Consolidation conduction abnormalities: 1%	Induction grade 3–4 LFT abnormalities: 44% Consolidation grade 3–4 LFT abnormalities: 2%
Estey et al. (2006) [19] non-randomized unicentric study (*n* = 44; HR: *n* = 19) Median age: 45 yrs	Induction ATRA 45 mg/m^2^ ATO 0.15 mg/kg/day (D10 of induction) + GO, IDA or GO + IDA (for HR) Consolidation ATO 0.15 mg/kg/d D1–5 W1–4, W9–12, W17–20, W25–28 + ATRA 45 mg/m^2^/d W1–2, W5–6, W9–10, W13–14, W17–18, W21–22, W25–26	- arrhythmias (*n* = 3) - QT prolongation (513 ms at end of induction) (*n* = 1) ATO discontinued, replaced with GO	- elevated LFT: 38.64% (no need to discontinue ATRA/ATO);
APL2012; Chen et al. (2021) [20] phase 3 multicentric RCT (*n* = 382) LR/IR APL (cons.) (*n* = 262): ATRA + ATO vs. ATRA + AC HR APL (cons.) (*n* = 129): ATRA + ATO + AC vs. ATRA + AC + ARA-C Median age: 38 yrs	Induction (847 pts) ATO 0.16 mg/kg/d (max. 10 mg) + ATRA0.25 mg/m^2^/d ± HU (LR) or IDA/DNR (IR or HR) Consolidation (ATO—382 pts) LR (2 cycles): ATO × 28 days ATRA × 14 days IR (3 cycles) ATRA × 14 days ATO × 28 days HR pts ATRA × 14 days IDA/DNR × 3 days ARA-C × 7 days (2 cycles) ATRA × 14 days ARA-C 1 g/m^2^/12 h, 3 days (1 cycle) Maintenance ATRA/ATO ± MTX nonHR: 3 cycles HR: 5 cycles	- QT prolongation: 4% (ATO) vs. 0.4% (CHT) (*p* < 0.0001); - grade 3–4 QT prolongation with no clinical impact (*n* = 2) (ATO);	- grade 1–2 DILI: 26.1% ATO vs. 17.9% CHT (*p* < 0.0001); - grade 3–4 DILI: 0.3% ATO vs. 0.5% CHT(*p* = 0.86);
Abaza et al. (2017) [21] (*n* = 187) HR (*n* = 54) and LR (*n* = 133) APL Median age: 50 yrs	Induction ATRA 0.15 mg/kg/d ATO 45 mg/m^2^/d Consolidation ATRA + ATO (4 cycles) + GO 9 mg/m^2^ on D1 for HR pts + LR pts if WBC > 10 × 10^9^/L during induction or IDA 12 mg/m^2^ if GO unavailable	QT prolongation: 7.5%	grade 3–4 DILI: 14%
Kayser et al. (2021) [22] prospective, real-world data non-HR APL (*n* = 154) Median age: 53 yrs	Induction ATRA 0.15 mg/kg/d + ATO 45 mg/m^2^/d (8 pts IDA 12 mg/m^2^; 2 pts ARA-C 100 mg/day, 2 days) Consolidation ATRA + ATO (4 cycles)	CV events (incl. QT prolongation): 3.90%	DILI: 13.64%

Legend: AC, anthracycline. APL, acute promyelocytic leukemia. ARA-C, cytarabine. ATO, arsenic trioxide. ATRA, all-trans retinoic acid. C, cycle/course. CHT, chemotherapy. Cons., consolidation. CV, cardiovascular. D, day. DILI, drug-induced liver toxicity. DNR, daunoribicin. GO, gemtuzumab ozogamicin. HR, high-risk APL. HU, hydroxyurea. IDA, idarubicin. incl., including. IR, intermediate risk. LFT, liver function tests. LR, low-risk APL. LV, left ventricle. MP, mercaptopurine. MTX, methotrexate. N, number. Pts, patients. RCT, randomized controlled/clinical trial. W, week. wks, weeks. yrs, years.

**Table 2 cancers-16-01160-t002:** Occurrence of gastrointestinal toxicity, and other adverse events reported in major clinical trials or real-world cohort studies based on treatment regimens for acute promyelocytic leukemia.

Study Details	Treatment Regimen	Other Toxicities
APL0406 (2013) [5] Platzbecker et al. (2016) [15] prospective phase 3 multicentric RCT nonHR APL ATRA + ATO (*n* = 77) vs. ATRA + CHT (*n* = 79) Median age: 44.6 yrs	Induction ATRA 0.15 mg/kg/d + ATO 45 mg/m^2^/d Consolidation ATRA + ATO (4 cycles)	Induction GI toxicity: 2% ATRA + ATO vs. 18.2% ATRA + CHT (*p* < 0.001) Hypertriglyceridemia: 22% ATRA + ATO vs. 22% ATRA + CHT (*p* = 0.76) Hypercholesterolemia: 10% ATRA + ATO vs. 8.7% ATRA + CHT (*p* = 0.55) Infections: 23% ATRA + ATO vs. 55% ATRA + CHT (*p* < 0.001) Consolidation Infections: 3% ATRA + ATO vs. 38% ATRA + CHT (*p* < 0.001) H1N1 viral pneumonia-related death (*n* = 1) with ATRA + ATO Hypercholesterolemia: 14% ATRA + ATO vs. 9% ATRA + CHT (*p* = 0.27)
AML17; Burnett et al. (2015) [14] phase 3 multicentric RCT ATRA/ATO ± GO (*n* = 116) vs. ATRA + CHT (*n* = 119) LR (*n* = 86) and HR (*n* = 30) APL Median age: 47 yrs	Induction ATRA 45 mg/m^2^/d + ATO 0.3 mg/kg D1–5 W1 and 0.25 mg/kg twice weekly W2–8 C1 ± GO 6 mg/m^2^ (HR pts + 7 non HR pts) Consolidation ATRA (2 wks on, 2 wks off schedule × 5 cycles) + ATO 0.3 mg/kg D1–5 W1 and 0.25 mg/kg twice weekly W2–4 of C2–5	Induction grade 3 elevated creatinine: 1% ATRA + ATO vs. 0% ATRA + CHT grade 3 diarrhea: 1% ATRA + ATO vs. 6% ATRA + CHT
APML4; Iland et al. (2012) [17,18] non-randomized phase 2 multicentric trial *n* = 124 pts (HR: *n* = 23) Median age: 44 yrs	Induction ATRA 45 mg/m^2^/d D1–36 + IDA 6–12 mg/m^2^ D2,4,6,8 + ATO 0.15 mg/kg D9–36 Consolidation (2 cycles) ATO + ATRA Maintenance (2 yrs) ATRA + 6-MP + MTX	Induction grade 3–4 skin reactions: 4%; - severe rash: *n* = 1 underwent induction only; grade 3–4 GI AE (nausea, vomiting, diarrhea, mucositis, enterocolitis): 28%; grade 3–4 infection:76% (*n* = 2 pts underwent induction only); Consolidation grade 3–4 dermatological complications: 1%; grade 3–4 GI toxicity: 3%; - infections: 3%.
APL2012; Chen et al. (2021) [20] phase 3 multicentric RCT (*n* = 382) LR/IR APL (cons.) (*n* = 262): ATRA + ATO vs. ATRA + AC HR APL (cons.) (*n* = 129): ATRA + ATO + AC vs. ATRA + AC + ARA-C Median age: 38 yrs	Induction (847 pts) ATO 0.16 mg/kg/d (max. 10 mg) + ATRA0.25 mg/m^2^/d ± HU (LR) or IDA/DNR (IR or HR) Consolidation (ATO—382 pts) LR (2 cycles): ATO × 28 days ATRA × 14 days IR (3 cycles) ATRA × 14 days ATO × 28 days HR pts ATRA × 14 days IDA/DNR × 3 days ARA-C × 7 days (2 cycles) ATRA × 14 days ARA-C 1 g/m^2^/12 h, 3 days (1 cycle) Maintenance ATRA/ATO ± MTX nonHR: 3 cycles HR: 5 cycles	rash: 8.3% (1st cons.) vs. 2.7% (3rd cons.) elevated creatinine: 2.3% (1st cons.) grade 3–4 hypertriglyceridemia: 13% (1st cons.) vs. 11.1% (3rd cons.) incidence of skin toxicity, elevated creatinine levels, and hypertriglyceridemia similar between the two pts subgroups; grade 3–4 infections more frequent in non-ATO arm, irrespective of APL risk
Abaza et al. (2017) [21] (*n* = 187) HR (*n* = 54) and LR (*n* = 133) APL Median age: 50 yrs	Induction ATRA 0.15 mg/kg/d ATO 45 mg/m^2^/d Consolidation ATRA + ATO (4 cycles) + GO 9 mg/m^2^ on D1 for HR pts + LR pts if WBC > 10 × 10^9^/L during induction or IDA 12 mg/m^2^ if GO unavailable	grade 3–4 infections: 23.5%
Kayser et al. (2021) [22] prospective, real-world data non-HR APL (*n* = 154) Median age: 53 yrs	Induction ATRA 0.15 mg/kg/d + ATO 45 mg/m^2^/d (8 pts IDA 12 mg/m^2^; 2 pts ARA-C 100 mg/day, 2 days) Consolidation ATRA + ATO (4 cycles)	Infections: 20.13%

Legend: AC, anthracycline. AE, adverse events. APL, acute promyelocytic leukemia. ARA-C, cytarabine. ATO, arsenic trioxide. ATRA, all-trans retinoic acid. C, cycle/course. CHT, chemotherapy. Cons., consolidation. D, day. DILI, drug-induced liver toxicity. DNR, daunoribicin. GI, gastrointestinal. GO, gemtuzumab ozogamicin. HR, high-risk APL. HU, hydroxyurea. IDA, idarubicin. incl., including. IR, intermediate risk. LR, low-risk APL. MP, mercaptopurine. MTX, methotrexate. N, number. Pts, patients. RCT, randomized controlled/clinical trial. W, week. wks, weeks. yrs, years.

**Table 3 cancers-16-01160-t003:** Occurrence of early death and differentiation syndrome in major clinical trials or real-world cohort studies based on treatment regimens for acute promyelocytic leukemia.

Study Details	Treatment Regimen	ED	DS
APL0406 (2013) [5] Platzbecker et al. (2016) [15] prospective phase 3 multicentric RCT nonHR APL ATRA + ATO (*n* = 77) vs. ATRA + CHT (*n* = 79) Median age: 44.6 yrs	Induction ATRA 0.15 mg/kg/d + ATO 45 mg/m^2^/d Consolidation ATRA + ATO (4 cycles)	ATRA + ATO: 0% vs. ATRA + CHT: 5.06%	Prophylaxis prednisone 0.5 mg/kg/d D1 to end of induction Incidence all grades: ATRA + ATO 19% vs. ATRA + CHT 16%, (*p* = 0.62) severe grade: ATRA + ATO6% vs. 6% ATRA + CHT (*p* = 0.99) Treatment ATRA, ATO, or both temporarily discontinued + DXM 10 mg/12 h 3 days minimum Mortality: 0%
AML17; Burnett et al. (2015) [14] phase 3 multicentric RCT ATRA/ATO ± GO (*n* = 116) vs. ATRA + CHT (*n* = 119) LR (*n* = 86) and HR (*n* = 30) APL Median age: 47 yrs	Induction ATRA 45 mg/m^2^/d + ATO 0.3 mg/kg D1–5 W1and 0.25 mg/kg twice weekly W2–8 C1 ± GO 6 mg/m^2^ (HR pts + 7 nonHR pts) Consolidation ATRA (2 wks on, 2 wks off schedule × 5 cycles) + ATO 0.3 mg/kg D1–5 W1 and 0.25 mg/kg twice weekly W2–4 of C2–5	similar 30-day mortality (*p* = 0.56) ATRA + ATO: 4% vs. ATRA + CHT: 6%	Prophylaxis none Incidence LR: 26.74%, HR: 23.33% Treatment DXM 10 mg/12 h ATRA/ATO temporarily discontinued Mortality: 0%
APML4; Iland et al. (2012) [17,18] non-randomized phase2 multicentric trial *n* = 124 pts (HR: *n* = 23) Median age: 44 yrs	Induction ATRA 45 mg/m^2^/d D1–36 + IDA 6–12 mg/m^2^ D2,4,6,8 + ATO 0.15 mg/kg D9–36 Consolidation (2 cycles) ATO + ATRA Maintenance (2 yrs) ATRA + 6-MP + MTX	ED: 3% causes: myocardial ischemia + cardiac arrest (*n* = 1), ICH (*n* = 2), cerebral edema (*n* = 1)	Prophylaxis prednisone 1 mg/kg/d PO (≥10 days) Incidence grade 3–4: 14% Mortality: 0%
Estey et al. (2006) [19] non-randomized unicentric study (*n* = 44; HR: *n* = 19) Median age: 45 yrs	Induction ATRA 45 mg/m^2^ ATO 0.15 mg/kg/day (D10 of induction) + GO, IDA or GO + IDA (for HR) Consolidation ATO 0.15 mg/kg/d D1–5 W1–4, W9–12, W17–20, W25–28 + ATRA 45 mg/m^2^/d W1–2, W5–6, W9–10, W13–14, W17–18, W21–22, W25–26	ED: 11.37% causes: ICH, pulmonary hemorrhage, stroke	Incidence 20% (LR: 12%; HR: 15.79%) Treatment methylprednisolone 45 mg/d × 7 days Mortality: 0%
APL2012; Chen et al. (2021) [20] phase 3 multicentric RCT (*n* = 382) LR/IR APL (cons.) (*n* = 262): ATRA + ATO vs. ATRA + AC HR APL (cons.) (*n* = 129): ATRA + ATO + AC vs. ATRA + AC + ARA-C Median age: 38 yrs	Induction (847 pts) ATO 0.16 mg/kg/d (max. 10 mg) + ATRA0.25 mg/m^2^/d ± HU (LR) or IDA/DNR (IR or HR) Consolidation (ATO—382 pts) LR (2 cycles): ATO × 28 days ATRA × 14 days IR (3 cycles) ATRA × 14 days ATO × 28 days HR pts ATRA × 14 days IDA/DNR × 3 days ARA-C × 7 days (2 cycles) ATRA × 14 days ARA-C 1 g/m^2^/12 h, 3 days (1 cycle) Maintenance ATRA/ATO ± MTX nonHR: 3 cycles HR: 5 cycles	ED: 4% causes: CNS and pulmonary hemorrhage (*n* = 20), severe infections (*n* = 8), stroke (*n* = 3), DS and pulmonary infections (*n* = 2), unknown (*n* = 1)	Treatment moderate DS or less: DXM 5–10 mg/d severe DS: ATRA/ATO discontinued + DXM ≤ 20 mg/d until WBC < 10.000/uL and no symptoms/signs of DS for 3 days Mortality: 0.24% (2/847 pts)
Abaza et al. (2017) [21] (*n* = 187) HR (*n* = 54) and LR (*n* = 133) APL Median age: 50 yrs	Induction ATRA 0.15 mg/kg/d ATO 45 mg/m^2^/d Consolidation ATRA + ATO (4 cycles) + GO 9 mg/m^2^ on D1 for HR pts + LRptsif WBC > 10 × 10^9^/L during induction or IDA 12 mg/m^2^ if GOunavailable	ED: 4% Causes: infections, hemorrhage, multiorgan failure	Prophylaxis methylprednisolone 50 mg/d × 5 days, then rapid tapering from D6 Incidence:11% Treatment: corticosteroids Mortality: 0%
Kayser et al. (2021) [22] prospective, real-world data non-HR APL (*n* = 154) Median age: 53 yrs	Induction ATRA 0.15 mg/kg/d + ATO 45 mg/m^2^/d (8 pts IDA 12 mg/m^2^; 2 pts ARA-C 100 mg/day, 2 days) Consolidation ATRA + ATO (4 cycles)	ED: 1% Causes: ARF (*n* = 1), ischemic cardiomyopathy (*n* = 1)	Prophylaxis prednisone 0.5 mg/kg/d D1 to end of induction Incidence: 4.55%

Legend: AC, anthracycline. APL, acute promyelocytic leukemia. ARF, acute respiratory failure. ARA-C, cytarabine. ATO, arsenic trioxide. ATRA, all-trans retinoic acid. C, cycle/course. CHT, chemotherapy. CNS, central nervous system. Cons., consolidation. D, day. DNR, daunoribicin. DS, differentiation syndrome. DXM, dexamethasone. ED, early death. GO, gemtuzumab ozogamicin. HR, high-risk APL. HU, hydroxyurea. ICH, intracerebral hemorrhage. IDA, idarubicin. IR, intermediate risk. LR, low-risk APL. MP, mercaptopurine. MTX, methotrexate. N, number. Pts, patients. RCT, randomized controlled/clinical trial. W, week. WBC, white blood cell(s) (count). wks, weeks. yrs, years.

**Table 4 cancers-16-01160-t004:** Neurological toxicities associated with different APL treatment regimens.

Author	Year	Country	APL pts	Treatment Regimen	Neurological Toxicities (incl. PTC)
Iland [18]	2015	Australia	124	Induction: ATRA + DNR + ATO Maintenance: ATRA + 6MP + MTX	Headache (*n* = 1, grade 3) during maintenance
Wang [47]	2011	Multicentric	348	ATRA + ATO vs. ATO	Headache 15.79% vs. 6.30%; RR = 1.96 95% CI 0.95–4.07
Wang [47]	2011	Multicentric	348	ATRA + ATO vs. ATO	Neuropathy 0% vs. 5.51%; RR = 0.32, 95% CI 0.04–2.24
Iland [17]	2012	Australia	124	ATRA + ATO + IDA	Grade 3/4 dizziness, mood alteration, musculoskeletal pain, or seizure Induction: 7 (6%) Consolidation 1:2 (2%) Consolidation 2:0 (0%)
Iland [17]	2012	Australia	124	ATRA + ATO + IDA	Grade 3/4 headache Induction: 4 (3%) Consolidation 1:2 (2%) Consolidation 2:0 (0%)
Soignet [11]	2001	USA	40	ATO in relapsed APL	Headache *n* = 24, 60%; grade 4: *n* = 1, 3% Insomnia *n* = 17, 43%; grade 4: *n* = 1, 3%
Platzbecker [15]	2017	Multicentric	276	ATRA + ATO vs. ATRA + chemotherapy (induction and consolidation)	Neurotoxicity rates (all grades; mainly reversible peripheral nerve neuropathy): – induction: similar rates (0.7% vs. 0%) – 1st consolidation: 4.2% vs. 0% (*p* = 0.02) – 2nd consolidation: 5% vs. 0% (*p* = 0.01) – 3rd consolidation: 5.9% vs. 0% (*p* = 0.006)
Estey [19]	2006	USA	25	ATRA + ATO vs. ATRA + chemotherapy	Peripheral neuropathy all grades (*n* = 3; 12%); severe (*n* = 1, 4%) (ATO discontinued, replaced with GO)
Loh [58]	2024	Australia	487	ATO-based therapy	Any grade neurotoxicity—23%: peripheral neuropathy:91% (grade 4: *n* = 1) encephalopathy:3% (grade 4: *n* = 2) seizures: 3% tremor: 2% ataxia: 3%

Legend: APL, acute promyelocytic leukemia. ARA-C, cytosine arabinoside (cytarabine). ATO, arsenic trioxide (intravenous administration if not otherwise specified). ATRA, all-trans retinoic acid. CI, confidence interval. DNR, daunorubicin. GO, gemtuzumab ozogamicin. IDA, idarubicin. MTX, methotrexate. PTC, pseudotumorcerebri. Pts, patients. RR, risk ratio. USA, United States of America. 6-MP, 6-mercaptopurine.

**Table 5 cancers-16-01160-t005:** Musculoskeletal toxicities associated with different APL treatment regimens.

Author	Year	Country	APL pts	Treatment Regimen	Musculoskeletal Toxicities
Wang [47]	2011	Multicentric	348	ATRA + ATO vs. ATO	Bone pain 5.85% vs. 1.57%; RR = 2.50, 95% CI 0.57–10.87
Efficace [59]	2021	Multicentric	162	ATRA + ATO vs. ATRA + chemotherapy	Less (overall) pain reported in ATRA + ATO subgroup
Efficace [59]	2021	Multicentric	161	ATRA + ATO (*n* = 83) vs. ATRA + chemotherapy (*n* = 78)	Long-term evaluation of comorbidities: Backpain: 30.1% vs. 28.2% Osteoarthritis/degenerative arthritis: 15.7% vs. 20.5%
Iland [17]	2012	Australia	124	ATRA + ATO + IDA	Grade 3/4 dizziness, mood alteration, musculoskeletal pain, or seizure – Induction: 7 (6%) – Consolidation 1:2 (2%) – Consolidation 2:0 (0%)

Legend: APL, acute promyelocytic leukemia. ATO, arsenic trioxide (intravenous administration if not otherwise specified). ATRA, all-trans retinoic acid. CI, confidence interval. IDA, idarubicin. RR, risk ratio.

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
