# Peer review of "Acute Promyelocytic Leukemia: Review of Complications Related to All-Trans Retinoic Acid and Arsenic Trioxide Therapy"

_cancers, 2024, doi:10.3390/cancers16061160_

Round 1

Reviewer 1 Report

Comments and Suggestions for Authors

Dear Editor,

Thank you very much for the opportunity to read the publication entitled
"Acute promyelocytic leukemia: a review of complications associated with
Atra and ATo therapy".
This manuscript is well written and i
n my opinion, it helps systematize
knowledge about the safety and effectiveness of the ATRA plus ATO scheme.
Nevertheless, there are some comments that I think need to be taken into
acounts by the authors:
1. Respect to QTc prolongation - I miss comment about drugs that may prolog
QTc
2. Could the authors indicate cleary infection profilaxis that should be
followed especially during induction phase
3. I do not like name of secondary malignancies in case of APL. In APL we
may have two type of tumours: MDS/AML or solid tumours. MDS/AML could be related
to chemotherapy, but in case of solid tumour we really do not know if it is
secondary or second independent tumour.
I miss also recommendations for oncological follow-up of APL patients after
finishing of APL therapy.
4. Should we recommend cardiological follow-up of APL patients treated with
ATRA plus ATO?
5. I think there is an error in "Infection complications" the incidence of
grade 3-4 infections wqs 14,1% with ATRA+chemotherapy vs 30,8% with ATRA+ATO??
Or 22,7% in ATRA-chemotherapy va 47,8% in the ATRAplus ATO??

Author Response

Thank you very much for the opportunity to read the publication entitled
"Acute promyelocytic leukemia: a review of complications associated with
Atra and ATo therapy".
This manuscript is well written and in my opinion, it helps systematize
knowledge about the safety and effectiveness of the ATRA plus ATO scheme.
Nevertheless, there are some comments that I think need to be taken into
acounts by the authors:
Response: We would like to thank you for your valuable comments which helped us improve the manuscript. All suggestions were taken into consideration and appropriate information, as well as required corrections, were provided. New/corrected parts are highlighted inyellow to facilitate the assessment of changes. We did our best to fulfil the expectations and we hope that you will be satisfied with our corrections.All in all, we thank you for your positive comments and appreciation regarding our manuscript.

  1. Respect to QTc prolongation - I miss comment about drugs that may prolog
    QTc

Response: Thank you for your suggestion. Our initial manuscript briefly mentioned the caution required regarding drug-drug interactions that could lead to QTc prolongation. Considering your advice, we have expanded this section to include specific examples of drug classes that could interact with ATO and result in prolonged QTc intervals: “The drugs most frequently associated with QT prolongation include antiemetics (ondansetron, prochlorperazine), antidepressants (including tricyclic antidepressants and selective serotonin reuptake inhibitors), antibiotics (quinolones), antifungals (azoles), potassium-wasting diuretics, and antiarrhythmics (amiodarone, sotalol). Azole antifungals, including itraconazole, voriconazole, and posaconazole, have been linked to QT prolongation. In contrast, isavuconazole is the only azole that does not influence QT prolongation. Furthermore, at certain plasma concentrations, isavuconazole has been demonstrated to have QT-shortening effects [66].”

  1. Could the authors indicate cleary infection profilaxis that should be
    followed especially during induction phase

Response: Thank you for your question. In our manuscript, we have incorporated the recommendations from the NCCN guidelines. The NCCN Guidelines for the Prevention and Treatment of Cancer-Related Infections indicate that patients with acute leukemia are at significant risk of infection. Therefore, during periods of neutropenia, the use of fluoroquinolones may be considered for antibacterial prophylaxis. Additionally, antifungal prophylaxis should be taken into account during neutropenia, with careful consideration of potential drug-drug interactions. Prophylaxis against HSV is recommended throughout all treatment cycles [74]. Currently, treatments involving ATO and ATRA are considered less immunosuppressive. In clinical practica, decisions on antibacterial and antifungal prophylaxis now largely depend on local protocols.

  1. I do not like name of secondary malignancies in case of APL. In APL we
    may have two type of tumours: MDS/AML or solid tumours. MDS/AML could be relatedto chemotherapy, but in case of solid tumour we really do not know if it is
    secondary or second independent tumour.I miss also recommendations for oncological follow-up of APL patients after
    finishing of APL therapy.

Response: We appreciate your recommendation. In response, we have made the necessary corrections within the manuscript and have discussed about t-MN, secondary neoplasia and second cancer. Furthermore, we have included a paragraph dedicated to the oncological monitoring of APL patients: At present, for the cancer follow-up of APL patients undergoing treatment with chemotherapy-free approaches, we recommend age-appropriate screening for potential new primary cancer together with comprehensive evaluations of patients’family and medical history, detailed physical examinations, and routine blood work.

  1. Should we recommend cardiological follow-up of APL patients treated with
    ATRA plus ATO?

Response: Thank you for your question. The reviewof the literature indicates that early cardiotoxic effects, particularly QT prolongation, are reversible and generally do not have a clinical impact. Regarding long-term cardiac events, there is a lack of data supporting the impact of ATO. Cardiological monitoring should be conducted based on symptoms and individual risk factors. We have added a paragraph to our paper stating: “At present, no specific recommendations have been issued regarding cardiovascular assessments during the long-term follow-up of APL patients who have received ATRA and ATO. Nevertheless, patients who have been prescribed anthracyclines and who exhibit multiple cardiovascular risk factors are considered at an elevated risk for cardiovascular dysfunction. From our point of view, the cases above should be monitored during APL long-term follow-up.”

  1. I think there is an error in "Infection complications" the incidence of
    grade 3-4 infections wqs 14,1% with ATRA+chemotherapy vs 30,8% with ATRA+ATO??Or 22,7% in ATRA-chemotherapy va 47,8% in the ATRAplus ATO??
    Response: Thank you for your comment. It was an error and we have made the necessary corrections accordingly.

Reviewer 2 Report

Comments and Suggestions for Authors

The purpose of this article is to review relevant literature data and discuss recommended management strategies for early and late nonhematologic complications that develop in patients diagnosed with APL and treated with ATO and ATRA regimens.

1. It is recommended to use the following relevant and useful sources in the article:

DOI: 10.1038/s41408-022-00753-y

DOI: 10.7314/apjcp.2015.16.13.5191

DOI: 10.3389/fonc.2021.762653

2. Consider a section about the use of ATO and ATRA in drug-resistant leukemia.

3. The manuscript should be updated with new sources from 2023-24.

4. Tables 3 and 4 are not mentioned in the text.

5. In the tables, there is no need to mention the author's name in the reference column and only the reference number is sufficient. In table number 1, only the reference number should be written in the corresponding column.

6. Regarding hematopoietic stem cell transplantation in leukemia and the effect of ATRA and arsenic trioxide, it is suggested to consider a section.

7. The manuscript should be completely checked in terms of English writing.

Comments on the Quality of English Language

Moderate editing of the English language is required.

Author Response

The purpose of this article is to review relevant literature data and discuss recommended management strategies for early and late nonhematologic complications that develop in patients diagnosed with APL and treated with ATO and ATRA regimens.

Response: We would like to thank you for your valuable comments which helped us improve the manuscript. All suggestions were taken into consideration and appropriate information, as well as required corrections, were provided. New/corrected parts are highlighted inyellow to facilitate the assessment of changes. We did our best to fulfil the expectations and we hope that you will be satisfied with our corrections.All in all, we thank you for your positive comments and appreciation regarding our manuscript.

  1. It is recommended to use the following relevant and useful sources in the article:

DOI: 10.1038/s41408-022-00753-y

DOI: 10.7314/apjcp.2015.16.13.5191

DOI: 10.3389/fonc.2021.762653

Response: We appreciate your suggestion. Accordingly, we have incorporated these three references into our manuscript.

  1. Consider a section about the use of ATO and ATRA in drug-resistant leukemia.

Response: Thank you for your comment, it is indeed an interesting suggestion, however, this is outside the current scope of our manuscript. We will take it into consideration when conducting further research in the field of APL.

  1. The manuscript should be updated with new sources from 2023-24.

Response: Thank you for your suggestion. We have updated our manuscript with references from 2023-2024.

  1. Tables 3 and 4 are not mentioned in the text.

Response: Thank you for your comment. We have reviewed the manuscript and incorporated Tables 3 and 4 into our text accordingly.

  1. In the tables, there is no need to mention the author's name in the reference column and only the reference number is sufficient. In table number 1, only the reference number should be written in the corresponding column.

Response: We appreciate your recommendation.Table number 1 has been divided into three separate tables to enhance comprehensibility and facilitate easier understanding.  In our opinion, listing the author's name in a table, along with the reference number, enhances the readability, utility, and academic integrity of a document, making it easier for readers to engage with and further investigate the content.

  1. Regarding hematopoietic stem cell transplantation in leukemia and the effect of ATRA and arsenic trioxide, it is suggested to consider a section.

Response:Thank you for your suggestion. The topic of allogeneic stem cell transplantation in APL treated with ATRA and ATO has been addressed by other colleagues in a separate article (https://doi.org/10.3390/cancers15164111 ). This article was published last year and is part of the same issue focusing on "Acute Promyelocytic Leukemia."

  1. The manuscript should be completely checked in terms of English writing.

Response: We appreciate your suggestion. The manuscript has been revised by a native speaker of the English language .

Reviewer 3 Report

Comments and Suggestions for Authors

The article "Acute Promyelocytic Leukemia: Review of Complications Related to ATRA and ATO therapy" bring a complete reprto fo APL complications related to ATRA and ATO therapy. The article is complete and exhaustive, but a review of the table 1  is necessary. The table 1 is of difficult comnprehesion and should redesigned in order to have a clear view of the data thhat could be added to the text

Author Response

The article "Acute Promyelocytic Leukemia: Review of Complications Related to ATRA and ATO therapy" bring a complete reprto fo APL complications related to ATRA and ATO therapy. The article is complete and exhaustive, but a review of the table 1  is necessary. The table 1 is of difficult comnprehesion and should redesigned in order to have a clear view of the data thhat could be added to the text

Response:Thank you for your suggestion regarding Table 1. We have revised the information from Table 1 and organized it into three separate tables for clarity and ease of understanding. The first table now includes information on cardiotoxicity and liver toxicity; the second table is dedicated to ED and DS; and the third table encompasses other toxicities.

Round 2

Reviewer 2 Report

Comments and Suggestions for Authors

Corrections have been made and the manuscript is acceptable.